# A Novel Small Molecular Prostaglandin Receptor EP4 Antagonist, L001, Suppresses Pancreatic Cancer Metastasis

**DOI:** 10.3390/molecules27041209

**Published:** 2022-02-11

**Authors:** Jiacheng He, Xianhua Lin, Fanhui Meng, Yumiao Zhao, Wei Wang, Yao Zhang, Xiaolei Chai, Ying Zhang, Weiwei Yu, Junjie Yang, Guichao Li, Xuekui Du, Hankun Zhang, Mingyao Liu, Weiqiang Lu

**Affiliations:** 1Shanghai Key Laboratory of Regulatory Biology, Institute of Biomedical Sciences and School of Life Sciences, East China Normal University, Shanghai 200241, China; jche1900@163.com (J.H.); lxh13850209674@163.com (X.L.); yumiaozhao1111@163.com (Y.Z.); 52191300031@stu.ecnu.edu.cn (W.W.); 18867137829@163.com (Y.Z.); xlchai@bio.ecnu.edu.cn (X.C.); yzhang@bio.ecnu.edu.cn (Y.Z.); yuerweiwei6@163.com (W.Y.); yangjj8@hspharm.com (J.Y.); hkzhang@bio.ecnu.edu.cn (H.Z.); 2Department of Gynecologic Oncology, The International Peace Maternity and Child Health Hospital, School of Medicine, Shanghai Jiao Tong University, Shanghai 200030, China; mengfanhui90@163.com; 3Department of Radiation Oncology, Fudan University Shanghai Cancer Center, 270 Dong’an Road, Shanghai 200032, China; Guichaoli11@fudan.edu.cn; 4Department of Respiratory Medicine, Ningbo No. 2 Hospital, No. 41, Northwestern Street, Ningbo 315099, China; duxuekui306@sina.com

**Keywords:** pancreatic cancer, metastasis, EP4, small molecule, YAP

## Abstract

Metastatic pancreatic cancer remains a major clinical challenge, emphasizing the urgent need for the exploitation of novel therapeutic approaches with superior response. In this study, we demonstrate that the aberrant activation of prostaglandin E_2_ (PGE_2_) receptor 4 (EP4) is a pro-metastatic signal in pancreatic cancer. To explore the therapeutic role of EP4 signaling, we developed a potent and selective EP4 antagonist L001 with single-nanomolar activity using a panel of cell functional assays. EP4 antagonism by L001 effectively repressed PGE_2_-elicited cell migration and the invasion of pancreatic cancer cells in a dose-dependent manner. Importantly, L001 alone or combined with the chemotherapy drug gemcitabine exhibited remarkably anti-metastasis activity in a pancreatic cancer hepatic metastasis model with excellent tolerability and safety. Mechanistically, EP4 blockade by L001 abrogated Yes-associated protein 1 (YAP)-driven pro-metastatic factor expression in pancreatic cancer cells. The suppression of YAP’s activity was also observed upon L001 treatment in vivo. Together, these findings support the notions that EP4–YAP signaling axis is a vital pro-metastatic pathway in pancreatic cancer and that EP4 inhibition with L001 may deliver a therapeutic benefit for patients with metastatic pancreatic cancer.

## 1. Introduction

Pancreatic cancer, one of the most lethal malignancies, is the leading cause of cancer-related death worldwide [1]. For decades, the improvement in the survival of pancreatic cancer patients has been very slow, partly due to the fact that more than 50% patients are diagnosed with metastatic disease at first presentation [2]. The 5-year survival rate of patients with localized pancreatic cancer is up to 30%, but those with distant metastatic pancreatic cancer have less than 3% [1,3]. The chemotherapeutic agent gemcitabine is a standard first-line regimen for patients with metastatic pancreatic cancer, whereas it can only increase median survival time by less than a year [4,5]. A combination chemotherapy FOLFIRINOX has produced a significant improvement in survival over gemcitabine, while high toxicity limits its clinical benefit [6,7]. Metastatic pancreatic cancer remains a major unresolved clinical problem, highlighting the urgent need for improved therapeutic strategies with superior efficacy and less toxicity.

Prostaglandin E_2_ (PGE_2_) is considered as a pro-inflammatory mediator and as a key pro-tumorigenic factor of many types of human malignancies [8,9]. PGE_2_ exerts its functions by interacting with four distinct prostaglandin E receptors (EP1, EP2, EP3, and EP4), which all belong to the G-protein coupled receptor (GPCR) family. Among these receptors, EP4 is highly associated with tumor malignant progression [10,11]. Notably, the genetic inactivation or biochemical blockade of EP4 is capable of blocking the metastasis progression of breast, prostate, and lung cancer [12,13]. Despite these extensive studies, the precise molecular mechanisms and therapeutic accessibility of PGE_2_–EP4 signaling in metastatic pancreatic cancer is not fully understood.

Yes-associated protein (YAP) is a major downstream effector of the Hippo pathway, which was originally identified as a critical regulator of organ expansion and tissue development in mammals [14]. In recent years, accumulated evidence has shown that dysfunctional YAP is highly associated with human malignancies, including pancreatic cancer [15,16]. YAP acts as an oncogenic transcription factor and is required for tumor initiation and the development of pancreatic cancer [17]. Furthermore, aberrant YAP activation leads to the highly metastatic characteristic of tumor cells, thus promoting pancreatic cancer invasion and metastasis [18]. Nevertheless, the upstream regulators of YAP in metastatic pancreatic cancer remain elusive.

In this work, we demonstrate that membrane receptor EP4 is a critical regulator of cell migration and the invasion of pancreatic cancer cells. L001 was identified as a highly potent and selective EP4 antagonist. The activation of the Hippo–YAP pathway is required for the pro-metastatic activity of EP4, which can be negated by L001. Moreover, the drug combination of L001 and chemotherapeutic agent synergistically inhibits the hepatic metastasis of pancreatic cancer cells. Together, these findings suggest that EP4 is a putative therapeutic target for the treatment of patients with metastatic pancreatic cancer.

## 2. Results

### 2.1. EP4 Is Essential for PGE_2_-Induced Pancreatic Cancer Cell Migration

PGE_2_ is a lipid inflammatory mediator with pronounced pro-tumorigenic capacity [19]. EP4 was reported to be a major functional receptor of PGE_2_ in cancers [20], while its role in metastatic pancreatic cancer is still not well defined. We aimed to detect the EP4 protein expression in a panel of pancreatic cancer cell lines using Western blotting with a specific antibody. A high level of EP4 protein expression was observed across these cell lines, especially in Pan02 and BxPC-3 (Figure 1a). Meanwhile, RT-PCR analysis demonstrated a similar expression pattern of EP4 mRNA levels (Figure 1b). We then detected PGE_2_-induced cell migration in Pan02 and BxPC-3 cells using wound healing and transwell assays. In the wound healing assay, these two cell lines were scratched and allowed to migrate for 24 h in the presence of various concentrations of PGE_2_. As shown in Appendix A, PGE_2_ treatment significantly promoted the motility of both cancer cell lines as a reflection of elevated wound closure, when compared with the vehicle group. Additionally, the results of the transwell migration assay also indicate that PGE_2_ acted as a potent chemoattractant for both Pan02 and BxPC-3 cells in a dose-dependent manner (Figure 1c–e). For instance, 0.1 μM PGE_2_ treatment can result in a 1.2-fold and 2.5-fold increase in migrated Pan02 and BxPC-3 cells, respectively.

To investigate the functional role of EP4, we next silenced EP4 by a lentiviral-based shRNA system in both Pan02 and BxPC-3 cells. A substantial decrease in the EP4 protein was observed in two distinct EP4–shRNA-treated groups, when compared with the control group (Appendix A). Notably, the knocking down of EP4 completely suppressed the PGE_2_-induced cell migration of both Pan02 and BxPC-3 cells in wound healing assay (Appendix A). The transwell assay further supported the anti-migration capacity of EP4 gene silencing (Figure 1f–h). The knocking down of EP4 had no effect on the proliferation of pancreatic cancer cells (Pan02 and BxPC-3) (Appendix A). Collectively, we can conclude that EP4 is a critical regulator of the PGE_2_-induced cell migration of pancreatic cancer cells.

### 2.2. PGE_2_–EP4 Signaling Activates Hippo–Yap Pathway in Pancreatic Cancer Cells

Aberrant activation of the Hippo–YAP pathway has been recognized in multiple types of human cancers, including pancreatic cancer. YAP overexpression predicts liver metastasis and poor prognosis of patients with pancreatic cancer [21]. We then examined whether PGE_2_–EP4 modulates the Hippo–YAP pathway in pancreatic cancer cells. We firstly observed that exposure of Pan02 and BxPC-3 cells to PGE_2_ significantly increased the level of intracellular YAP protein over time in Western blotting analysis (Appendix A). YAP was the transcriptional co-activator for TEAD protein to activate downstream transcription [22,23,24]. Thus, a TEAD-driven luciferase reporter was used to quantify YAP activation in pancreatic cancer cells. As shown in Figure 2a,b, PGE_2_ markedly promoted TEAD-driven luciferase activity in both Pan02 and BxPC-3 cells with a max luminescence fold increase of 1.87 (*p* < 0.0001) and 1.56 (*p* = 0.0027), respectively. In addition, the expression of YAP target genes CYR61 and CTGF [24] was elevated upon PGE_2_ stimulation over time (Appendix A). Together, these results demonstrate that PGE_2_ activates the Hippo–YAP pathway in pancreatic cancer cells.

Next, we examined the role of EP4 in the PGE_2_-activated Hippo–YAP pathway. As shown in Figure 2c,d, the exogenous overexpression of EP4 induced a 2.62- and 5.54-fold increase in the TEAD reporter assay in Pan02 and BxPC-3 cells, respectively. In contrast, the knocking down of EP4 results in a 55.1% (*p* = 0.0018) and 13.6% (*p* = 0.034) reduction in luciferase activities in both Pan02 and BxPC-3 cells, respectively, when compared with the control cells. Accordingly, the mRNA expression of YAP as well as the YAP target genes CYR61 and CTGF was increased in EP4-overexpressing cells but decreased in EP4-knockdown cells (Figure 2e,f). Together, these findings suggest that PGE_2_–EP4 augmented the activation of the pro-metastatic Hippo–YAP pathway in pancreatic cancer cells.

### 2.3. L001 Is a Potent and Selective Antagonist of EP4 Receptor

To explore the therapeutic role of EP4, a carboxamido-benzoic acid derivative, L001 (Figure 3a), was identified as a novel EP4 antagonist through an extensive medicinal chemistry campaign (see Section 4). The antagonistic activities of L001 were evaluated by a panel of EP4 functional assays. EP4 couples with Gαs protein, which activates the adenylyl cyclase and stimulates the generation of the second messenger cAMP [25]. In the cAMP responsive element (CRE) luciferase assay using HEK293 cells, we found that L001 can dose-dependently inhibit PGE_2_-induced CRE-driven luciferase expression with an IC_50_ value of 7.29 ± 0.64 nM (Figure 3b). Of note, L001 is superior to E7046 in CRE reporter assay (IC_50_ = 40.6 ± 14.8 nM), a known EP4 antagonist evaluated in early clinical trials (Figure 3c) [26]. EP4 is capable of recruiting β-arrestin upon PGE_2_ treatment [27]. To assess the effect of L001 on β-arrestin recruitment, we used a stable and sensitive TANGO (transcriptional activation following arrestin translocation) assay [28] for EP4. An exogenous transcription factor tTA was fused with EP4 at its cytoplasmic tail and β-arrestin was tagged with TEV-protease (Figure 3d). PGE_2_ can significantly increase luciferase expression, while L001 observably blocked PGE_2_-induced EP4/β-arrestin2 interaction with an IC_50_ value of 0.16 ± 0.03 nM in TANGO assay (Figure 3e). Furthermore, ERK1/2 is an important downstream signaling effector of EP4 [29]. Western blotting analysis shows that L001 pre-treatment significantly blocked the PGE_2_-induced phosphorylation of ERK1/2 in a dose-dependent manner (Figure 3f).

We also established a general calcium flux assay [30] for prostaglandin E receptors and found that L001 exhibits a >600-fold greater selectivity in human EP4 than the other three prostaglandin E receptors (IC_50_ > 10 µM for EP1, EP2 and EP3; Figure 3g). Moreover, L001 was a potent EP4 antagonist in human (Figure 3g), monkey (Figure 3h), mouse (Figure 3i) and rat (Figure 3j) as detected by the calcium flux assay with IC_50_ values of 1.47 ± 0.02 nM, 5.24 ± 1.16 nM, 3.20 ± 0.28 nM and 14.25 ± 0.88 nM, respectively. Collectively, these observations indicate that L001 is a highly potent and selective EP4 antagonist.

### 2.4. L001 Impairs Pancreatic Cancer Migration and Invasion In Vitro

Next, we investigated whether the EP4 antagonist L001 could affect PGE_2_-induced pancreatic cancer cell migration using the wound healing assay and transwell assay. Scratched Pan02 and BxPC-3 cells were treated with various concentrations of L001 in the presence of 0.1 μM PGE_2_ and allowed to migrate for 24 h. As shown in Appendix A, PGE_2_-triggered wound closure was significantly blocked by L001 in a dose-dependent manner in both cell lines. Similarly, reduced migrated cells in the lower layer of the transwell were observed upon L001 treatment (Figure 4a–c). These biochemical blockade results, combined with aforementioned genetic knockdown data highlight the critical role of EP4 in the regulation of the cell migration of pancreatic cancer cells.

Previously, studies have reported that the PGE_2_–EP4 signaling axis can also promote cancer cell invasion [31]. We thus exploited a three-dimensional (3D) tumor spheroid invasion assay to assess the effect of L001 on cancer cell invasion. Both Pan02 and BxPC-3 cell lines were encapsulated with a layer of 100% Matrigel and covered with growth medium containing 10% Matrigel. Spheroid invasion was monitored at day 7. We found that both cell lines exhibited an obvious elongated, fibroblastoid shape and displayed filopodium-like protrusions as a very extended morphology (black arrow) in the presence of 0.1 μM PGE_2_ (Figure 4d). Of note, L001 treatment completely retracted PGE_2_-induced membrane surface projections and restored a similar morphology to the control group (Figure 4d). Furthermore, in line with EP4 knockdown results, EP4 inhibition by L001 did not affect the cell viabilities of both Pan02 and BxPC-3 cells (Figure 4e). Collectively, these data suggest that EP4 antagonist L001 impairs cell migration and the invasion of pancreatic cancer cells.

### 2.5. L001 Abrogated Hippo–YAP Activation and Pro-Metastatic Factors Expression in Pancreatic Cancer Cells

We then aimed to evaluate the effect of L001 on Hippo–YAP activation in Pan02 and BxPC-3 cells. We found that L001 treatment remarkably suppressed PGE_2_-driven YAP/TEAD reporter activities in pancreatic cancer cells. Notably, L001 was better than E7046 in the reporter assay under the same concentration (Figure 5a,b). Moreover, in comparison with E7046, L001 treatments also resulted in stronger reductions in the mRNA expression levels of YAP as well as YAP target genes CYR61 and CTGF induced by PGE_2_ (Figure 5c,d).

Epithelial–mesenchymal transition (EMT) enables cancer cells to perform dynamic changes to obtain strong mobilities and to shape into an enhanced aggressive metastatic phenotype by regulating EMT-associated markers [32]. Tumor cells tend to promote the expression levels of mesenchymal-related markers including Vimentin and Snail, as well as suppress the expression of epithelial-related genes including E-cadherin [33]. Previous research has argued that YAP could regulate the expression of EMT markers in pancreatic cancer cells [34]. Hence, we investigated whether EP4 antagonism would affect the expression of EMT markers. As expected, L001 remarkably abrogated the PGE_2_-aroused upregulation of mRNA levels of Vimentin and Snail, while it reversed the PGE_2_-mediated downregulation of E-cadherin’s expression, suggesting an impaired EMT capacity (Figure 5e–g). Together, these findings suggest that L001 is capable of impairing the PGE_2_–EP4-driven activation of the pro-metastatic Hippo–YAP pathway in pancreatic cancer cells.

### 2.6. L001 Treatment Impairs Hepatic Metastasis of Pancreatic Cancer In Vivo

Next, we established a liver metastases model in C57BL/6 mice to assess the in vivo effect of L001 on metastatic pancreatic cancer. Luciferase-labeled Pan02 cells (Pan02-Luc) were intrasplenically injected and tumor progression was monitored via bioluminescence imaging every 7 days. We noticed that signals from the thoracic cavity in the vehicle control group were gradually diffused and enlarged with a median tumor volume quantified as total bioluminescence flux displayed a 51.4-fold increase from day 0 to day 28 (Figure 6a,b). Oral administration of L001 (75 mg/kg/day) significantly promotes Pan02 tumor regression with a reduction in median luminescence signal of 80.57%, when compared with the vehicle control group (*p* < 0.05) on day 28. The clinical drug E7046 (75 mg/kg/day) results in a 69.13% loss of luminescence signal in comparison with the vehicle control group, while no statistical differences were observed (*p* > 0.05) (Figure 6a,b). Gemcitabine is a first-line chemotherapeutic agent for metastatic pancreatic cancer. Notably, we observed that the drug combination of L001 and gemcitabine had a greater potency on tumor growth and diffusion (99.06% reduction in bioluminescence compared with control group, *p* < 0.01) than either the single-agent treatment of L001 or gemcitabine (91.79% reduction of bioluminescence compared with control group, *p* < 0.05) on day 28 (Figure 6a,b).

In order to catch a virtual degree of hepatic metastases, mouse livers were separated and the corresponding bioluminescence was recorded. Our data show that Pan02 cell liver metastases were reduced by 76.42%, 85.14% (*p* = 0.0002, *n* = 6), 93.2% (*p* < 0.0001, *n* = 6) and 97.87% (*p* < 0.0001, *n* = 6) upon treatment with E7046, L001, gemcitabine, and L001/gemcitabine combination, respectively (Figure 6c,d). Of note, H&E imaging of the liver sections revealed remarkably reduced metastatic clusters surrounded by dense peritumoral stroma following single-agent treatment of E7046, L001 or gemcitabine, and no actual tumor was observed upon L001/gemcitabine co-treatment (Figure 6e). In line with in vitro observation, we found that treatment with L001 resulted in decreased protein levels of YAP and reduced the mRNA expressions of its downstream pro-metastatic genes of Vimentin and Snail in hepatic metastasis loci (Appendix A).

Importantly, the combination of L001 and gemcitabine remarkably prolonged tumor-bearing mouse survival: the control group (*n* = 9; median survival = 34 days), L001 group (*n* = 9; median survival = 42 days), E7046 group (*n* = 9; median survival = 35 days), gemcitabine group (*n* = 9; median survival = 46 days), and combination group (*n* = 9; median survival = 58 days) (*p* < 0.0001, Log-rank test; Figure 6f). Body weight loss was barely found among L001, gemcitabine and their combination groups, indicating these treatments were well tolerated in vivo (Figure 6g). Collectively, these findings suggest that L001 treatment could have a striking anti-tumoral effect and a strong synergism with gemcitabine for metastatic pancreatic cancer.

## 3. Discussion

Metastatic pancreatic cancer remains an almost universally lethal malignancy with dismal prognosis [35]. The liver is the most common metastatic organ of pancreatic cancer [36]. Liver metastasis of pancreatic cancer exhibits three main fatal characteristics, including small tumors with advanced metastases, rapid relapse after radical resection and extremely short survival rate, of within only 3–6 months [37,38]. The current first-line chemotherapies, including gemcitabine or FOLFIRINOX, have limited benefits for metastatic pancreatic cancer, with a median overall survival of less than a year [7]. Hence, effective agents with superior therapeutic response and less side effects are urgently needed. In this work, we focused on the therapeutic intervention of malignant metastatic programs in pancreatic cancer by targeting the EP4–YAP signaling pathway. We thus developed a small-molecule EP4 antagonist L001 with high potency and selectivity. EP4 blockade by L001 repressed the activation of pro-metastatic Hippo–YAP pathway, and facilitated the anti-cancer efficacy of gemcitabine with good tolerance and safety in a preclinical model of metastatic pancreatic cancer.

PGE_2_ is a bioactive lipid mediator with profound pro-tumorigenic activities [39]. Tumors can directly or indirectly upregulate the secretion of PGE_2_ and subsequently activate the corresponding EP receptors for intracellular signaling transduction. The EP4 receptor is closely associated with features of malignant cells, including invasion, migration and metastasis [29]. Thus, pharmacologically blocking the EP4 receptor is considered as an attractive strategy for the treatment of metastatic cancers. However, the therapeutic potential and underlying mechanism of EP4 in metastatic pancreatic cancer has not been well defined. Our findings demonstrate that the EP4–YAP signaling axis is a critical pro-metastatic pathway in pancreatic cancer and that pharmacological blockade of EP4 may deliver a therapeutic benefit for patients with metastatic pancreatic cancer.

The Hippo pathway is closely associated with the migration and invasion of pancreatic cancer cells [40,41]. Studies have reported that YAP, the major downstream effector of the Hippo pathway, is overexpressed in clinical tumor samples derived from pancreatic cancer patients and is a prognostic marker for metastatic pancreatic cancer [42,43]. Given the pharmacological inaccessibility of transcription coactivator YAP, we then turn to the upstream regulator of the pro-metastatic Hippo–YAP pathway. Previous studies have established membrane-bound GPCRs as critical modulators of the Hippo–YAP pathway [44]. The EP4 receptor, as a member of the Class A GPCR superfamily, might have the potential to mediate YAP signaling. As expected, we demonstrated that Hippo–YAP is activated by the EP4 receptor and is involved in EP4-driven metastatic program in pancreatic cancer. Although the detailed regulation mechanism of EP4–YAP signaling was not fully understood, our results provide a clear rationale for exploring EP4 blockade to target the aberrant activation of YAP in pancreatic cancer and other human diseases in which YAP has a pathological role.

Our and other groups have reported a panel of EP4 antagonists in the past years [30,45,46,47]. Small molecule L001 represents a next-generation EP4 antagonist harboring a functional carboxamido-benzoic acid skeleton. Notably, the introduction of trifluoromethyl as a substituent group on the benzene ring brings superior antagonistic activities against EP4. Specifically, L001 exhibited an over 5-fold better antagonism efficacy on EP4 when compared with clinical drug E7046, in both calcium flux assay and CRE reporter assay. Importantly, the in vivo therapeutic effects of L001 were also better than E7046 in terms of blocking hepatic metastasis and prolonging lifespan in a Pan02 metastatic pancreatic cancer model. These results indicate a potential capacity and a promising prospect for L001 to turn into a clinical candidate compound for the treatment of patients with metastatic pancreatic cancer.

Gemcitabine has been the cornerstone of neo-adjuvant, adjuvant and palliative therapies for patients with metastatic pancreatic cancer for the last couple of decades, as no alternative therapeutic options exist [48]. Even so, the response to gemcitabine in patients with advanced pancreatic cancer is poor, with barely any reduction in cancer metastasis, mainly owing to rapid drug elimination, high hydrophilicity and the propensity to elicit drug resistance, notorious hallmarks of metastatic pancreatic cancer [49]. Evidently, the anti-tumor and especially anti-metastasis potentials of gemcitabine on pancreatic cancer metastasis should be harnessed for improvement. A drug combination strategy provides more effective therapeutic outcomes and delivers efficacy previously unachievable with monotherapies [50]. Since we determined the profound pro-metastatic properties of EP4 in pancreatic cancer, we speculated that the combining of our novel EP4 antagonist L001 and gemcitabine might offer a more effective therapeutic regimen for metastatic pancreatic cancer. Indeed, we observed that the drug combination of L001 and gemcitabine demonstrated a cooperative anti-metastasis activity with almost complete regressions of hepatic metastases in the Pan02 metastatic model. These results elaborate the potential of L001 to overcome the chemoresistance of patients with metastatic pancreatic cancer. Of note, our results also pave the way to explore whether EP4 antagonism has a synergistic anti-cancer effect with other treatment modalities in metastatic pancreatic cancer [51].

## 4. Materials and Methods

### 4.1. Experimental Framework

The experimental framework is shown in Figure 7.

### 4.2. Cell Culture

The pancreatic cancer cell lines (Pan02, MIA PaCa-2, BxPC-3, PANC-1, Panc-28, SW1990, Capan-2) were obtained from ATCC (Manassas, VA, USA). CHO and HEK293 were purchased from the National Infrastructure of Cell Line Resource (Shanghai, China). HEK293, Pan02, BxPC-3, PANC-1 and Panc-28 were cultured in Dulbecco’s modified Eagle’s medium (DMEM) supplemented with 10% fetal bovine serum (FBS) with 100 U/mL of penicillin and 100 µg/mL of streptomycin (Gibco, Carlsbad, CA, USA). MIA PaCa-2 was grown in DMEM supplemented with 10% FBS and 2.5% horse serum (Gibco, Carlsbad, CA, USA). SW1990 was maintained in L15 medium supplemented with 10% FBS. Capan-2 was cultured in RPMI-1640 (Gibco, Carlsbad, CA, USA) supplemented with 10% FBS. CHO was grown in DMEM-F12 (Gibco, Carlsbad, CA, USA) supplemented with 10% FBS. All cells were maintained at 37 °C in a humidified 5% CO_2_ atmosphere. Cell lines were identified by short tandem repeat analysis and checked for mycoplasma contamination.

### 4.3. Cell Viability Assay

Cells were seeded into 96-well plates and treated with varied concentrations of compounds for 72 hrs. Cell viabilities were determined by using MTS reagent (Promega, Madison, WI, USA) according to the manufacturer’s manual. Spectrophotometric absorbance at 490 nm was measured by Flexstation^®^ 3 Multi-Mode Microplate Reader (Molecular Devices, San Jose, CA, USA). The experiments were independently repeated three times.

### 4.4. Wound Healing Assay

The wound healing assay was performed according to a standard protocol as described before [52]. Briefly, Pan02 and BxPC-3 cells were grown as a confluent monolayer in 12-well plates (Corning, Corning, NY, USA) overnight and starved with serum-free medium for 6 h. The monolayer cells were disrupted with a P100 micropipette tip (JET BIOFIL, Guangzhou, China) and rinsed with PBS to remove detached cells. Medium containing different concentrations of compounds was added subsequently. After 24 h, treated cells were stained with 2 μM calcein-AM (Yeasen, Shanghai, China) for 20 min and washed with PBS. Photos were taken of the area in the scraped zone of each dish at 100× magnification by using an inverted fluorescent microscope (OLYMPUS, Tokyo, Japan).

### 4.5. Transwell Assay

The transwell assay was conducted as described previously [53], using 8 μm pore transwell inserts (Corning, NY, USA). Cells were suspended in serum-free DMEM and seeded into upper chambers at appropriate density. The bottom chamber was filled with DMEM supplemented with 10% fetal bovine serum and indicated concentrations of PGE_2_. The compounds were added in both upper and bottom chambers. After 24 h, the cells were fixed in 4% paraformaldehyde and stained with 0.1% crystal violet (Sangon Biotech, Shanghai, China). Migrated cells were counted and quantified using image J.

### 4.6. 3D Matrigel Invasion Assay

The 3D Matrigel invasion assay was performed as described [54]. Pan02 and BxPC-3 cells were plated above a layer of Matrigel (ThermoFisher, Shanghai, China) and covered with growth medium containing 10% Matrigel. The culture medium containing different concentrations of compounds was changed every two days. After 7 days, cells with stellate invasive structure were photographed (OLYMPUS, Tokyo, Japan).

### 4.7. Real-Time qPCR

Total RNA was extracted from cells using TRIzol reagent. The cDNA was synthesized by RT-PCR kit (Vazyme Biotech, Nanjing, China). The band intensity was determined by the gel image analysis system (Bio-Rad, Hercules, CA, USA). SYBR Green-based qPCR (Yeason, Shanghai, China) was conducted subsequently. Relative mRNA levels of target genes were normalized with β-actin. The sequences of primers are listed in Appendix A. The raw data are listed in Appendix A.

### 4.8. Immunoblotting

Cells were lysed in RIPA lysis buffer with phosphatase and protease inhibitors (Calbiochem, San Diego, CA, USA) and denatured by heating at 100 °C for 10 min. Proteins were separated on 10% bis-tris polyacrylamide gels, transferred to nitrocellulose filter membranes (Millipore, Burlington, NJ, USA), and blocked with 5% BSA (Sangon Biotech, Shanghai, China). The membranes were incubated with the indicated primary antibodies overnight and subsequently probed with indicated secondary antibodies. Images of the blots were taken and analyzed by Odyssey imaging system (Li-COR Biosciences, Lincoln, NE, USA). The primary antibodies were listed as follows: Human-YAP (1:10,000 dilution, CY5381, abways); Mouse-YAP (1:1000 dilution, ab205270, abcam); EP4 (1:500 dilution, 101775, Cayman Chemical); GAPDH (1:10,000 dilution, Ab181602, abcam).

### 4.9. RNA Interference

The doxymycin-inducible shRNA expression vector pSingle-tTS-shRNA was used for RNA interference. shRNA specific to human or mouse Ptger4 was prepared by Synbio Technologies, Suzhou, China. Cells were transfected with corresponding shRNA vectors by lipofectamine 2000 reagent (Thermo-Invitrogen, Shanghai, China), and the medium was replaced by DMEM with 10% fetal bovine serum and 3 μM doxymycin (Sigma, River Edge, NJ, USA) after 5 h. EP4 protein expression was determined by immunoblotting using indicated primary antibodies. The sequences of shRNA are listed in Appendix A.

### 4.10. Luciferase Reporter Assay

CHO-K1 cells or pancreatic cancer cell lines were transfected with a cAMP responsive element (CRE) reporter or TEAD reporter using lipofectamine 2000 according to the manufacturer’s manual. Transfected cells were treated with varied concentrations of compounds for 24 h. Then, cells were lysed and luciferase activities were determined using Dual-Luciferase Reporter Assay Kit (Promega, Madison, WI, USA) [55].

### 4.11. TANGO Assay

TANGO assay was performed as described previously with some modification [28]. Briefly, CHO-K1 cells were co-transfected with plasmids of EP4-TANGO, β-arrestin-TEV and tTA-luc at a ratio of 0.5:1:1. After 4 h, transfected cells were seeded into 96-well plates and cultured overnight. Then, cells were starved for 5 h and treated with different concentrations of compounds. After incubation for 16 h, cells were lysed and the luciferase activities were determined using Dual-Luciferase Reporter Assay Kit (Promega, Madison, WI, USA).

### 4.12. Calcium Flux Assay

Calcium flux assay was conducted as described in our previous study [30]. Briefly, CHO-Gα16 cells were transfected, respectively, with human EP1, human EP2, human EP3, human EP4, monkey EP4, mouse EP4 and rat EP4 overexpression constructs using lipofectamine 2000 according to the manufacturer’s manual. Cells were then seeded into a 96-well black plate with a cell density of 2 × 10^4^ cells/100 μL per well for a 16-h culture. On the next day, cells were incubated for 45 min with reagents (100 μL/well) of Calcium-5 Assay Kit (Molecular Devices, San Jose, CA, USA). Then, cells were pre-treated with indicated concentrations of L001 for 15 min and stimulated with PGE_2_ (EC_80_). The calcium flux was detected subsequently by a Flexstation^®^ 3 Multi-Mode Microplate Reader (Molecular Devices, San Jose, CA, USA).

### 4.13. Liver Metastasis Animal Model and Bioluminescent Imaging

Male C57/BL6 mice (6–8 weeks) were obtained from the National Rodent Laboratory Animal Resources (NRLAR, Shanghai, China). All mice were housed according to Institutional Animal Care and Use Committee guidelines and under a protocol approved by East China Normal University Ethics Committee with respect to animal care and welfare assurance. The bioluminescent liver metastasis mouse model was established as described previously [56]. In brief, 2 × 10^6^ Pan02/luc cells suspended in 50 μL of PBS were injected into mice spleen. Then, all of the mice were randomly assigned to five groups: control group (CMC-Na, p.o. daily), L001 group (75 mg/kg, p.o. daily), E7046 group (75 mg/kg, p.o. daily), gemcitabine group (25 mg/kg, i.p. twice a week), and L001 + gemcitabine group (75 mg/kg, po; daily + 25 mg/kg, i.p. twice a week). Tumor metastasis was monitored by using an in vivo imaging system following intraperitoneal injection with D-Luciferin (2 mg per mouse; Promega). Bioluminescence intensity was recorded every seven days by a Xenogen IVIS-200 Optical in vivo imaging system (PerkinElmer; Waltham, MA, USA) for up to 4 weeks. The mouse body weight was measured every other day. Survival curve analysis was performed using GraphPad Prism 7.

### 4.14. Immunohistochemistry Analysis

Mouse liver tissues were harvested, fixed, paraffin embedded, and sectioned (8 μm). The tissues were permeabilized with 0.25% TritonX-100 at room temperature for 15 min, blocked with 1% FBS in PBS, and followed by incubation in primary antibodies at 4 °C overnight: rabbit anti-YAP (1:50 dilution, ab205270, abcam). The sections were then washed with PBS, incubated with anti-rabbit secondary antibody for 2 h and dyed with DAPI (10 μg/mL in PBS) for 5 min at room temperature. Slides were mounted using DPX Mountant (Electron Microscopy Sciences) and images were obtained by inverted fluorescence microscope (Olympus, Tokyo, Japan).

### 4.15. Chemistry

#### 4.15.1. Preparation of L001

All chemicals were purchased from Adamasbeta Ltd., J&K Scientific Ltd., Sigma-Aldrich Inc., or Aladdin-Reagents Inc., and solvents were used as received from Tansoole or Sigma-Aldrich Inc. without further purification. All reactions were executed with standard procedures. The progress of each reaction was monitored by TLC on thin-layer plates. ^1^H NMR and ^13^C NMR spectra were generated on a Bruker 400 or 500 MHz instrument and obtained as DMSO-d_6_ solutions. NMR chemical shifts were reported in δ (ppm) using the δ 2.50 signal of DMSOd_6_ (^1^H NMR), and the δ 39.50 signal of DMSO-d_6_ (^13^C NMR) as the reference standards. The purity of all final compounds (≥95%) was established by analytical HPLC, which was carried out on an Agilent 1200 HPLC system with an ACE Excel 5 C18 column (5 μm, 4.6 × 250 mm), column temperature 40 °C, with detection at 254 or 280 nm on a variable wavelength detector G1314B. Compound **1** was reacted with elemental sulfur and ethyl cyanoacetate through Gewald reaction in ethanol solution to obtain **2**, which then underwent bromination reaction to generate **3**, and then underwent Barbier nucleophilic addition reaction with 3-(trifluoromethyl)benzaldehyde at −78 °C to generate the desired product, **4**. Then, **4** reacted with excess triethylsilane and trifluoroacetic acid in dichloromethane to remove the hydroxyl group to obtain **5**. Finally, compound **8** was obtained through hydrolysis and the HATU coupling reaction (Figure 8a). The ^13^C NMR of L001 was shown in Figure 8b.

#### 4.15.2. Reagents and Conditions

(a) Ethyl cyanoacetate, S, morpholine, EtOH, 60 °C, 12 h, 94–97%; (b) tert-butyl nitrite, CuBr2, Dioxane, MeCN, 0 °C to rt, 3 h, 58–62%; (c) 3-(trifluoromethyl)benzaldehyde, 2.5 M n-BuLi in hexane, Et2O, −78 °C to rt, 8 h, 60–67%; (d) TFA, Et3SiH, CH2Cl2, 0 °C, 2 h, 50–55%; (e) LiOH·H2O, THF/MeOH/H2O = 2:2:1, 70 °C, 3 h, 96–98%; (f) methyl-4-[(1S)-1-aminoethyl]benzoate, HATU, DIPEA, rt, 5 h, 80–85%.

### 4.16. Statistical Analysis

All data were analyzed by using GraphPad Prism 7. Experiments were performed in three biological replicates with at least two technical replicates and data were presented as mean ± standard error of mean (SEM). The null hypothesis is that the means of the measured values were identical in compared groups. If the result shows a statistically significant change in the compared groups, the null hypothesis is rejected. When comparing two groups, statistical significance was analyzed by unpaired Student’s t test. When comparing multiple groups, one-way ANOVA with the multiple comparisons test was conducted. Survival analysis was performed as Kaplan–Meier curves and analyzed by the log-rank test. A *p* value of < 0.05 was considered statistically significant (* *p* < 0.05; ** *p* < 0.01; *** *p* < 0.001).

## 5. Conclusions

Metastatic pancreatic cancer is one of the most lethal malignancies worldwide. Patients can hardly benefit from traditional therapeutic regimens, emphasizing the urgent need to develop novel therapeutic approaches with better response. Our current study illustrated that PGE_2_–EP4 signaling is capable of activating the pro-metastatic Hippo–YAP pathway in pancreatic cancer in vitro and in vivo. Moreover, we developed a novel and potent EP4 antagonist L001, which impaired the EP4-mediated activation of the pro-metastatic Hippo–YAP pathway and synergistically acted with gemcitabine to target hepatic metastasis of pancreatic cancer. The comprehensive preclinical findings showed here indicates hat L001 is a promising clinical candidate for the treatment of patients with metastatic pancreatic cancer through targeting the EP4–YAP axis.

## Figures and Tables

**Figure 1 molecules-27-01209-f001:**
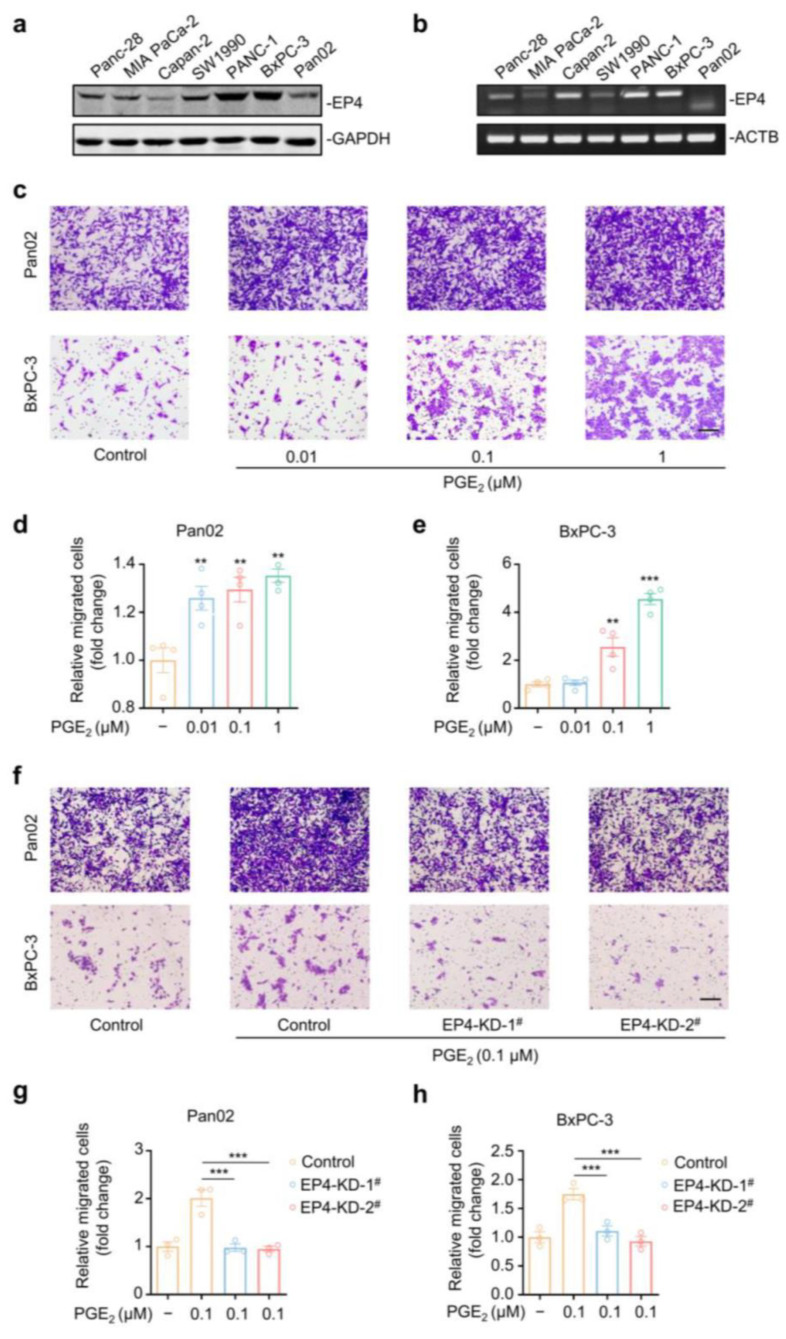
EP4 is essential for PGE_2_-induced pancreatic cancer cell migration. (**a**) Immunodetection of EP4 protein expression in six human pancreatic cancer cell lines and a murine pancreatic cancer cell line, Pan02. GAPDH was used as a loading marker. (**b**) RT-PCR analysis of mRNA levels of EP4 in the above seven pancreatic cancer cell lines. ACTB was used as a loading marker. (**c**) Cell migration analysis of Pan02 and BxPC-3 cells treated with indicating concentrations of PGE_2_ for 16 h via transwell assay. Scale bar, 200 μm. (**d**,**e**) Quantification of cell migration of Pan02 (**d**) and BxPC-3 (**e**) cells treated with indicating concentrations of PGE_2_ as in (**c**) (*n* = 3). (**f**) Cell migration analysis of Pan02 and BxPC-3 cells transfected with a scramble shRNA (Control) or shRNAs targeting EP4 (EP4-KD), followed by a 24-h treatment with DMSO or 0.1 μM PGE_2_ via transwell assay. Scale bar, 200 μm. (**g**,**h**) Quantification of cell migration of Pan02 (g) and BxPC-3 (**h**) cells in (**f**) (*n* = 4). The *p* value was calculated by one-way ANOVA with multiple comparison test. ** indicates *p* < 0.01; *** indicates *p* < 0.001. All data are presented as mean ± SEM.

**Figure 2 molecules-27-01209-f002:**
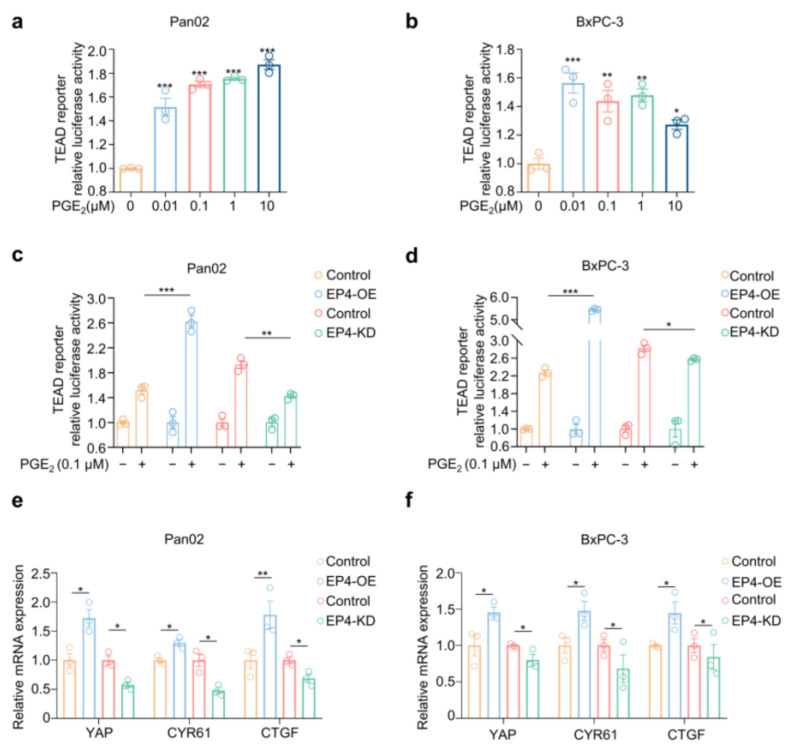
PGE_2_–EP4 signaling activates Hippo–YAP pathway in pancreatic cancer cells. (**a**,**b**) Relative TEAD4 transactivation activity in Pan02 (**a**) and BxPC-3 (**b**) cells treated with DMSO or indicated concentrations of PGE_2_ for 16 h (*n* = 3). (**c**,**d**) Relative TEAD4 transactivation activity in Pan02 (**c**) and BxPC-3 (**d**) cells transfected with a EP4 over-expressing plasmid (EP4-OE) or an empty vector (Control) as well as a shRNA targeting EP4 (EP4-KD) or a scramble shRNA (Control), followed by treatment with DMSO or 0.1 μM PGE_2_ for 16 h (*n* = 3). (**e**,**f**) Relative mRNA expressions of YAP, CYR61 and CTGF in Pan02 (**e**) and BxPC-3 (**f**) cells transfected with an EP4 over-expressing plasmid (EP4-OE) or an empty vector (Control) as well as a shRNA targeting *Ptger4*/*PTGER4* (EP4-KD) or a scramble shRNA (Control), followed by treatment with DMSO or 0.1 μM PGE_2_ for 12 h. mRNA levels were determined by qPCR (*n* = 3). The *p* value was calculated by one-way ANOVA with multiple comparison test. * indicates *p* < 0.05; ** indicates *p* < 0.01; *** indicates *p* < 0.001. All data are presented as mean ± SEM.

**Figure 3 molecules-27-01209-f003:**
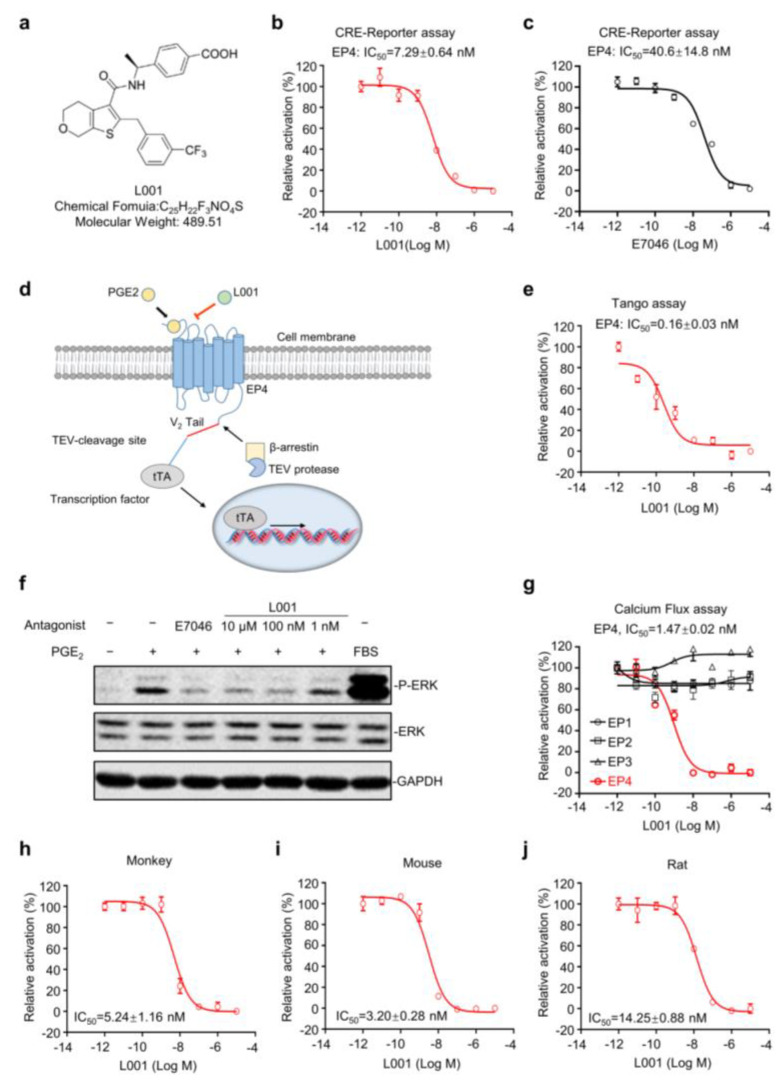
L001 is a potent and selective antagonist of EP4. (**a**) Chemical structure of L001. (**b**,**c**) Dose–response curve of L001 (**b**) or E7046 (**c**) against EP4 in HEK293 cells in CRE reporter assay (*n* = 3). (**d**) Schematic diagram of TANGO assay. (**e**) Dose–response curve of L001 against EP4 in CHO-K1 cells via TANGO assay (*n* = 3). (**f**) Immunodetection of ERK1/2 phosphorylation. Serum-starved cells were pre-incubated with indicating concentrations of L001 or E7046 for 20 min and then stimulated with DMSO or 30 nM PGE_2_ subsequently for 5 min. Fetal bovine serum (FBS) was used as a positive control. GAPDH was used as a loading marker. (**g**) Dose–response curves of L001 on human EP1, EP2, EP3 and EP4 receptors in calcium flux assay (*n* = 3). (**h**–**j**) Dose–response curves of L001 on monkey (**h**), mouse (**i**) and rat (**j**) EP4 receptors in calcium flux assay (*n* = 3). All data are presented as mean ± SEM of three independent experiments.

**Figure 4 molecules-27-01209-f004:**
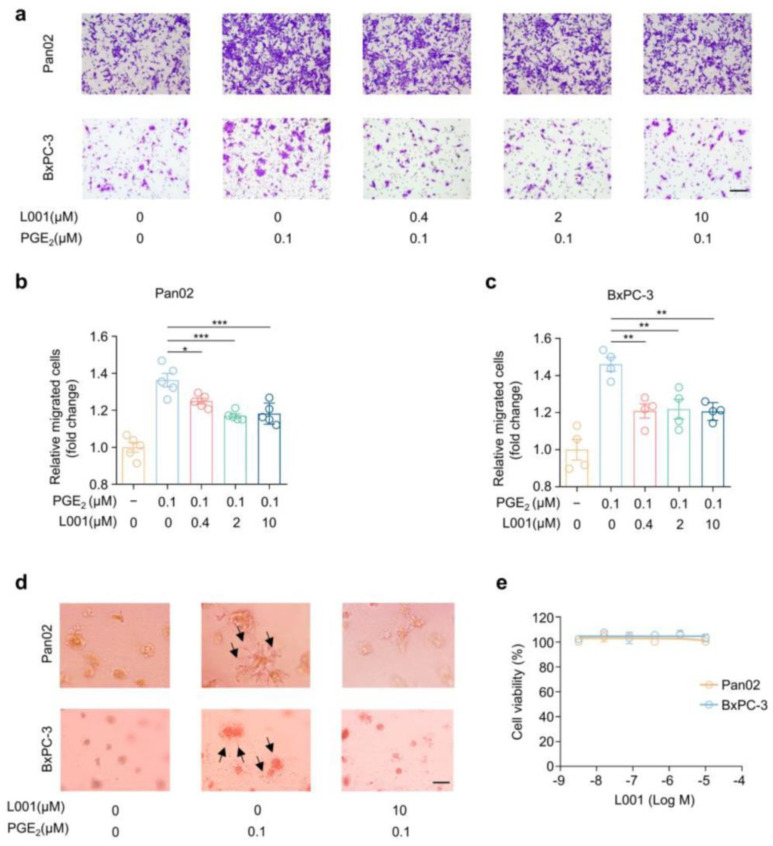
L001 impairs pancreatic cancer migration and invasion in vitro. (**a**) Cell migration analysis of Pan02 and BxPC-3 cells treated with indicated concentrations of L001 and PGE_2_ for 24 h via transwell assay. Scale bar, 200 μm. (**b**,**c**) Quantification of cell migration of Pan02 (**b**) and BxPC-3 (**c**) cells in (**a**) (*n* = 3). (**d**) Representative cellular morphology images of Pan02 and BxPC-3 cells treated with indicated concentrations of L001 and PGE_2_. Filopodium-like protrusions are pointed to by black arrows. (**e**) Cell viability curves of Pan02 and BxPC-3 cells treated with indicating concentrations of E7046 for 72 h. The *p* value was calculated by one-way ANOVA with multiple comparison test. * indicates *p* < 0.05; ** indicates *p* < 0.01; *** indicates *p* < 0.001. All data are presented as mean ± SEM.

**Figure 5 molecules-27-01209-f005:**
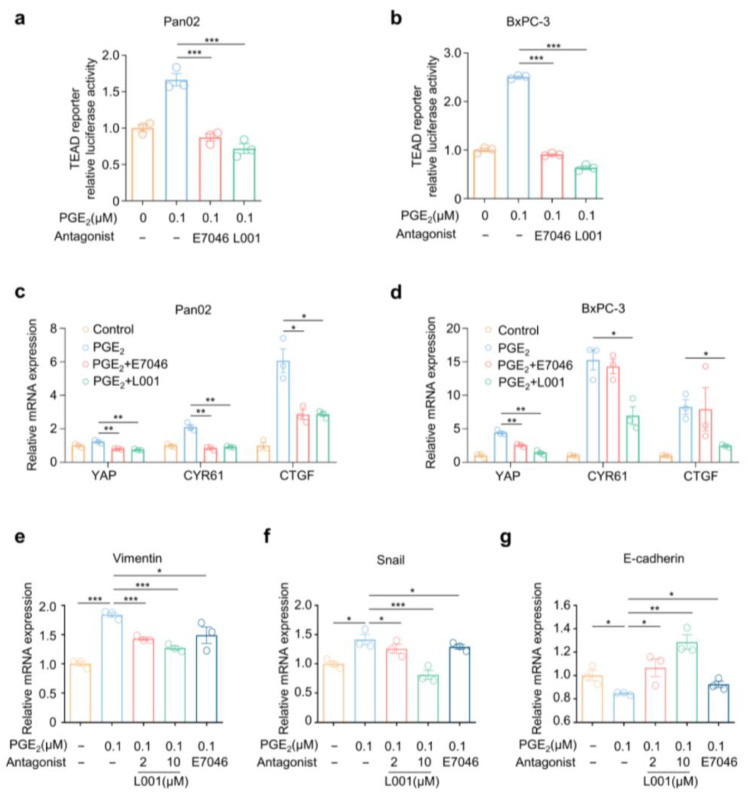
L001 abrogated Hippo–YAP activation and pro-metastatic factors expression in pancreatic cancer cells. (**a**,**b**) Relative TEAD4 transactivation activity in Pan02 (**a**) and BxPC-3 (**b**) cells treated with indicated concentrations of PGE_2_, E7046 and L001 for 16 h (*n* = 3). (**c**,**d**) Relative mRNA expressions of YAP, CYR61 and CTGF in Pan02 (**c**) and BxPC-3 (**d**) cells treated with indicated PGE_2_ (0.1 μM), E7046 (10 μM) and L001 (10 μM) for 12 h. mRNA levels were determined by qPCR (*n* = 3). (**e**–**g**) Relative mRNA expressions of Vimentin (**e**), Snail (**f**) and E-cadherin (**g**) in BxPC-3 cells treated with indicating concentrations of PGE_2_, E7046 and L001 for 12 h. mRNA levels were determined by qPCR (*n* = 3). The *p* value was calculated by one-way ANOVA with multiple comparison test. * indicates *p* < 0.05; ** indicates *p* < 0.01; *** indicates *p* < 0.001. All data are presented as mean ± SEM.

**Figure 6 molecules-27-01209-f006:**
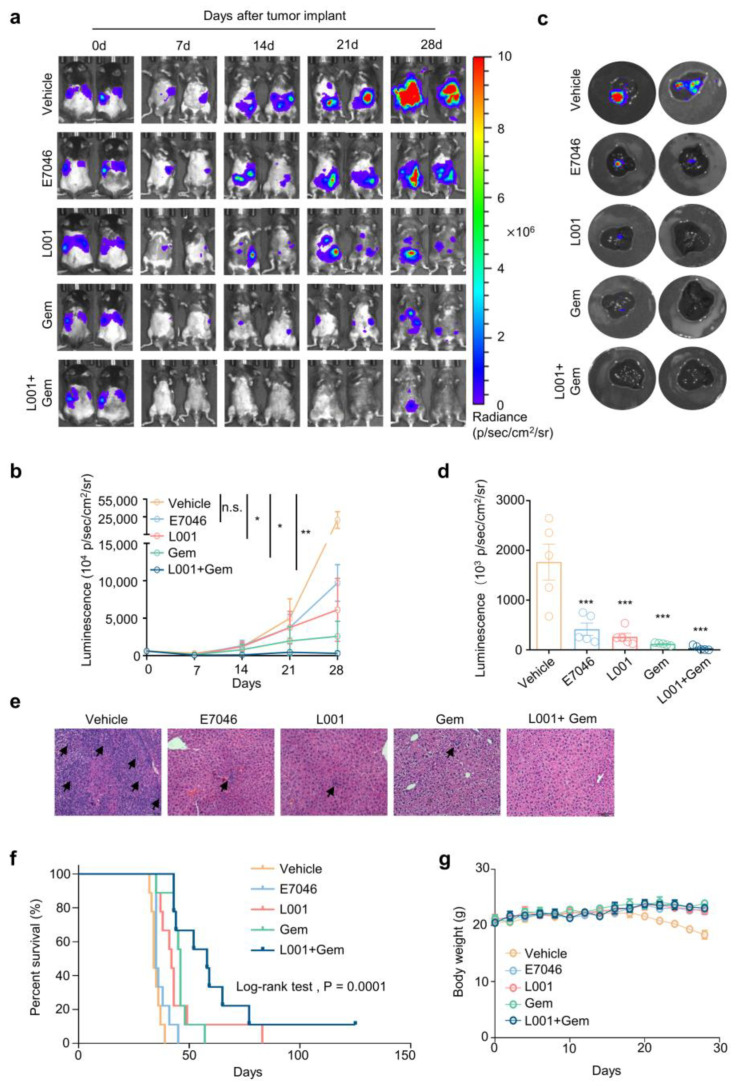
L001 treatment impairs hepatic metastasis of pancreatic cancer in vivo. (**a**,**b**) Representative bioluminescence images (**a**) and corresponding quantitative growth curves (**b**) of Pan02-Luc metastasis model upon indicated treatments (*n* = 6). (**c**,**d**) Representative bioluminescence images (**c**) and corresponding bioluminescence quantification (**d**) of murine livers from indicated groups of Pan02-Luc metastasis model (*n* = 6). (**e**) Representative HE staining of murine liver tissues from indicated groups of Pan02-Luc metastasis model. (**f**) Event-free survival of mice from Pan02-Luc metastasis model upon indicated treatments. Differences in survival were analyzed using a Kaplan–Meier log-rank test (*n* = 9). (**g**) Body weights of mice upon indicated treatments (*n* = 6). The *p* value was calculated by one-way ANOVA with multiple comparison test. * indicates *p* < 0.05; ** indicates *p* < 0.01; *** indicates *p* < 0.001. All data are presented as mean ± SEM.

**Figure 7 molecules-27-01209-f007:**
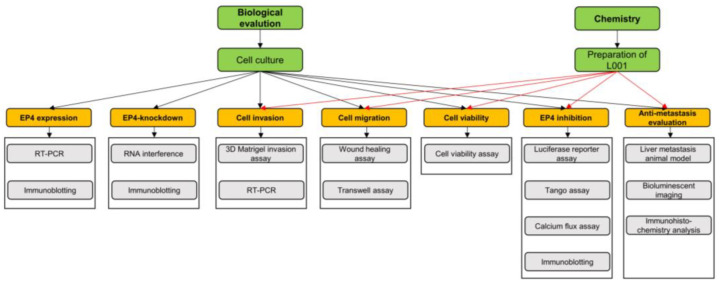
Framework of experiments. The framework figure illustrating the experiments used in this paper.

**Figure 8 molecules-27-01209-f008:**
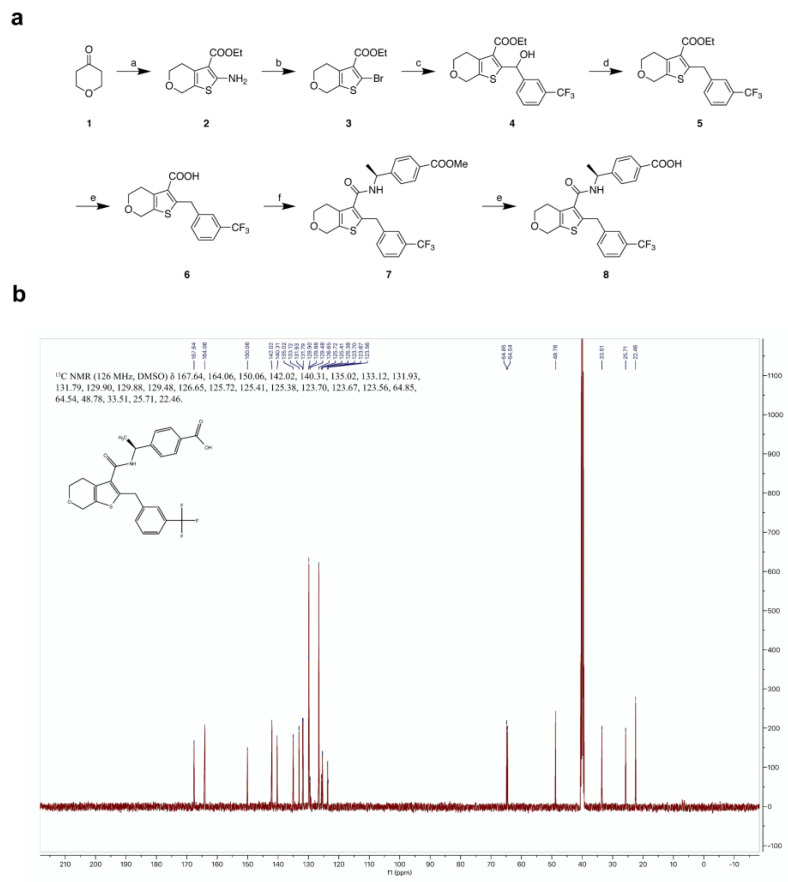
The synthesis and the ^13^C NMR of L001. (**a**) The synthesis of L001. (**b**) The ^13^C NMR of L001.

## Data Availability

Not applicable.

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
