# Peer review of "A Novel Small Molecular Prostaglandin Receptor EP4 Antagonist, L001, Suppresses Pancreatic Cancer Metastasis"

_molecules, 2022, doi:10.3390/molecules27041209_

Round 1

Reviewer 1 Report

The authors presented a manuscript entitled “A Novel Small Molecular Prostaglandin Receptor EP4 Antagonist, L001, Suppresses Pancreatic Cancer Metastasis” in which the authors elucidate on a possible drug target and a possible pathway that may be linked to pancreatic cancer metastasis.

The article is scientifically sound and well designed. The English syntax and grammatic is pretty good as well, it was perfectly understandable, so I believe no further corrections, rather than a final proof reading, are necessary. The authors made an accurate statistical analysis, that is well described in the methodology, only lacking in the depiction of the null hypothesis, which is common practice, even so, I believe the null hypothesis should always be clearly stated. The findings are very interesting and seem relevant for the elucidation of this specific cancer metastasis. Moreover, as this is not my primary field of research, I can’t attest for the soundness of the employed assays; however, the discussion was clear enough to be easily understood and I believe that the present manuscript is relevant, as every new finding in cancer pathways are very useful to better tackle these difficult illnesses that cancers present.

L 512-517 The Statistical analysis is well explained, however, despite the standard use of similarity between groups as the null hypothesis, the authors should clearly state which was their null hypothesis in this section

Author Response

Point 1: The authors presented a manuscript entitled “A Novel Small Molecular Prostaglandin Receptor EP4 Antagonist, L001, Suppresses Pancreatic Cancer Metastasis” in which the authors elucidate on a possible drug target and a possible pathway that may be linked to pancreatic cancer metastasis.

The article is scientifically sound and well designed. The English syntax and grammatic is pretty good as well, it was perfectly understandable, so I believe no further corrections, rather than a final proof reading, are necessary. The authors made an accurate statistical analysis, that is well described in the methodology, only lacking in the depiction of the null hypothesis, which is common practice, even so, I believe the null hypothesis should always be clearly stated. The findings are very interesting and seem relevant for the elucidation of this specific cancer metastasis. Moreover, as this is not my primary field of research, I can’t attest for the soundness of the employed assays; however, the discussion was clear enough to be easily understood and I believe that the present manuscript is relevant, as every new finding in cancer pathways are very useful to better tackle these difficult illnesses that cancers present.

Response 1: We thank the Reviewer for these positive comments.

Point 2: L 512-517 The Statistical analysis is well explained, however, despite the standard use of similarity between groups as the null hypothesis, the authors should clearly state which was their null hypothesis in this section.

Response 2: We have modified the null hypothesis statements in the statistical analysis section of the revised manuscript.

Reviewer 2 Report

The review article entitled " A Novel Small Molecular Prostaglandin Receptor EP4 Anta-go-nist, L001, Suppresses Pancreatic Cancer Metastasis". The Jiacheng He et al., demonstrate that aberrant activation of prostaglandin E2 (PGE2) receptor 4 (EP4) is a pro-metastatic signaling in pancreatic cancer. The authors have explored the therapeutic role of EP4 signaling, and they have developed a potent and selective EP4 antagonist L001 with single-nanomolar activity using a panel of cell functional assays. These findings states that that EP4/YAP signaling axis is a vital pro-metastatic pathway in pancreatic cancer and that EP4 inhibition with L001 may deliver a therapeutic benefit for patients with metastatic pancreatic cancer.

The manuscript comprises all the necessary elements of scientific novelty. The manuscript is well written and substantiated with detailed analysis. Authors should carryout the following points during revision. 

Line 70: Check the format of PGE2 and throughout the manuscript.

The methods section should contain a framework figure of listing all the analysis/steps done in this paper.

Authors should provide the Ct values (qPCR’s raw data) in the supplementary for scrutiny.

Authors should state the “Experiments were done in three biological replicates with at least two technical replicates and data were presented as…… in statistical analysis section.

Separate conclusions section should be provided.

Authors must concentrate on the formatting, and use of symbols, etc., There is no uniformity was observed in few places of the manuscript.

Discussion and conclusion section looks shallow. It needs to be improved.  Discuss more and it will be useful to the readers for ease of understanding.

Author Response

Point 1: The review article entitled " A Novel Small Molecular Prostaglandin Receptor EP4 Anta-go-nist, L001, Suppresses Pancreatic Cancer Metastasis". The Jiacheng He et al., demonstrate that aberrant activation of prostaglandin E2 (PGE2) receptor 4 (EP4) is a pro-metastatic signaling in pancreatic cancer. The authors have explored the therapeutic role of EP4 signaling, and they have developed a potent and selective EP4 antagonist L001 with single-nanomolar activity using a panel of cell functional assays. These findings states that that EP4/YAP signaling axis is a vital pro-metastatic pathway in pancreatic cancer and that EP4 inhibition with L001 may deliver a therapeutic benefit for patients with metastatic pancreatic cancer.

The manuscript comprises all the necessary elements of scientific novelty. The manuscript is well written and substantiated with detailed analysis. Authors should carryout the following points during revision. 

Response 1: We thank the Reviewer for these positive comments.

Point 2: Line 70: Check the format of PGE2 and throughout the manuscript.

Response 2: We have unified the format of PGE2 throughout the manuscript.

Point 3: The methods section should contain a framework figure of listing all the analysis/steps done in this paper.

Response 3: We have added an experimetial framework figure in Materials and Methods section of the revised manuscript.

Point 4: Authors should provide the Ct values (qPCR’s raw data) in the supplementary for scrutiny.

Response 4: We have added Ct values in the appendix of the revised manuscript.

Point 5: Authors should state the “Experiments were done in three biological replicates with at least two technical replicates and data were presented as…… in statistical analysis section.

Response 5: We have added this statement in statistical analysis section of the revised manuscript.

Point 6: Separate conclusions section should be provided.

Response 6: We have added a separate conclusion section in the revised manuscript.

Point 7: Authors must concentrate on the formatting, and use of symbols, etc., There is no uniformity was observed in few places of the manuscript.

Response 7: We thank the Reviewer for this comment and we have carefully revised our manuscript to ensure the uniformity of the formatting, symbols and terms in the revised manuscript.

Point 8: Discussion and conclusion section looks shallow. It needs to be improved. Discuss more and it will be useful to the readers for ease of understanding.

Response 8: We thank the Reviewer for this comment and we have improved our disccsion section in the revised manuscript.